# Wastewater Sequencing—An Innovative Method for Variant Monitoring of SARS-CoV-2 in Populations

**DOI:** 10.3390/ijerph19159749

**Published:** 2022-08-08

**Authors:** Michal Tamáš, Alena Potocarova, Barbora Konecna, Ľubos Klucar, Tomas Mackulak

**Affiliations:** 1Department of Environmental Engineering, Institute of Chemical and Environmental Engineering, Faculty of Chemical and Food Technology, Slovak University of Technology, Radinského 9, 81237 Bratislava, Slovakia; 2Institute of Physiology, Faculty of Medicine, Comenius University in Bratislava, 81372 Bratislava, Slovakia; 3Institute of Molecular Biomedicine, Faculty of Medicine, Comenius University in Bratislava, Sasinkova 4, 81108 Bratislava, Slovakia; 4Institute of Molecular Biology, Slovak Academy of Sciences, Dúbravská Cesta 21, 84551 Bratislava, Slovakia

**Keywords:** COVID-19, sampling, RNA extraction, RT-qPCR, mutations

## Abstract

The SARS-CoV-2 outbreak has already affected more than 555 million people, and 6.3 million people have died. Due to its high infectivity, it is crucial to track SARS-CoV-2 outbreaks early to prevent the spread of infection. Wastewater monitoring appears to be a powerful and effective tool for managing epidemiological situations. Due to emerging mutations of SARS-CoV-2, there is a need to monitor mutations in order to control the pandemic. Since the sequencing of randomly chosen individuals is time-consuming and expensive, sequencing of wastewater plays an important role in revealing the dynamics of infection in a population. The sampling method used is a crucial factor and significantly impacts the results. Wastewater can be collected as a grab sample or as a 24 h composite sample. Another essential factor is the sample volume, as is the method of transport used. This review discusses different pretreatment procedures and RNA extraction, which may be performed using various methods, such as column-based extraction, TRIzol, or magnetic extraction. Each of the methods has its advantages and disadvantages, which are described accordingly. RT-qPCR is a procedure that confirms the presence of SARS-CoV-2 genes before sequencing. This review provides an overview of currently used methods for preparing wastewater samples, from sampling to sequencing.

## 1. Introduction

Low infection doses, the capability of rapid mutations, robustness against several disinfection approaches, and in many cases lack of successful treatment are the main concerns regarding environmental viruses. Genetic drifts/shifts in influenza have been responsible for several epidemics or pandemics in which millions of people have died [1]. Nowadays, we can observe similar scenarios in the case of other viruses, such as coronaviruses. The current SARS-CoV-2 virus causing COVID-19 disease imposed significant global health and economic burdens. There were over 555,000,000 cases, with approximately 6.3 million deaths as of 18th January 2022.

SARS-CoV-2 was also identified in human feces [2]. The virus infects angiotensin-converting enzyme 2 (ACE2)-expressing cells abundant in the gastrointestinal tract’s small intestine, although the respiratory tract is the major location of infection. The virus is shed through feces and is predominantly noninfectious after exposure to gastric juices [3]. Shedding of the virus to stools probably occurs soon after infection and the virus could therefore be present in wastewater [2,4]. This may happen a week before an infected person evolves symptoms, goes to a testing facility, undergoes nasopharyngeal testing, and tests positive in lab analysis [5,6].

In October 2020, Larsen and Wigginton [7] reported that, due to limited testing capacity, large numbers of persons with mild or asymptomatic disease, and hospitalizations delayed by weeks, wastewater monitoring could be a powerful tool to manage SARS-CoV-2 epidemiological situations in particular geographical areas. This was proven by several scientific teams worldwide that began surveillance of wastewater, and their results were used as complementary data to support health authorities in decision making related to public health restrictions [8,9,10,11,12,13,14]. Wastewater-based epidemiology offers several benefits in comparison to mass testing of populations. It is a cost-effective way to survey pandemic situations of large communities or small, closed communities [15,16,17]. According to the newest research, it displays infection dynamics before standard diagnostic testing. The primary molecular technique used for testing is real-time polymerase chain reaction (RT-qPCR), which is also important during PCR-based sequencing, as in testing for SARS-CoV-2. With emerging mutations in the viral genomes, sequencing a sufficient number of clinical samples is another important asset in controlling the pandemic. Unfortunately, the method is often expensive, even after cost optimization. Here again, sequencing of wastewater, considered as a pooled specimen of many individuals, could alleviate the economic burden of the method and have similar benefits to those mentioned before in relation to RT-qPCR [18]. A sequenced wastewater sample gives us information about a sufficiently large portion of the population, while results from clinical sequencing are from one individual. It was proven that sequencing of nasopharyngeal swabs and wastewater in the same area results in similar metagenomic profiles [19]. Additionally, several variants of SARS-CoV-2 were detected by wastewater sequencing that were not captured by clinical sequencing [19]. In this minireview, we would like to point out the state of the art of the process from sampling to wastewater sequencing for SARS-CoV-2 and its use in the research into and prediction of new variants. All the steps discussed are visualized in Figure 1 for a better understanding of the described methodologies.

## 2. Methodology

All the presented literature was researched on PubMed with the keywords SARS-CoV-2 OR COVID-19 AND wastewater OR sewage AND sequencing and their combinations. Additionally, few preprints were selected for this review. Table 1 summarizes all the available published literature regarding wastewater pretreatment, processing, and analysis of SARS-CoV-2 presence. It lists separate scientific groups and their approaches to sampling, pre-treatment, nucleic acid extraction, RT-qPCR methodology, and the creation sequencing libraries and sequencing technology options. The following text discusses each step more fully, intending to critically point out the pros and cons of the different methodologies.

### 2.1. Samples and Sampling Methods

Correct sampling is a crucial step in the whole analysis. The gold standard is a 24 h composite sampling that displays a good representation of daily load [49]. The volume of the collected samples varies from 100 mL to 2000 mL between the research groups, with 500 mL as the most preferred sample volume [20,24,30,34,37,40]. Viral ribonucleic acid (RNA), which is prone to degradation, is analyzed in the portion of the sample right away, where the rest is ideally stored at 4–5 °C for possible later reanalysis [26,28,30,35,36,43,50]. At this temperature, the virus and its nucleic acids are stable for approximately one week and should be stored at lower temperatures afterward [51]. Therefore, the majority of groups decided to transport the sample refrigerated or on ice to inhibit RNases [20,21,27,29,30,31,32,33,36,37,38,39,40,41,44,45]. The majority of the studies included in the review used 24 h composite sampling of influent. Grab samples were adopted by eleven groups, with volumes ranging from 200 to 1000 mL [20,25,30,33,34,40,41,44,46,48,52]. Grab sampling is fast but unreliable, as it captures only a defined time point, and the use should be considered based on sampling facility and expected outcome, e.g., during morning hygiene in hospital [49]. In Bi et al.’s study, samples were collected twice per day and mixed afterwards with a final volume ranging from 300 to 500 mL to mimic composite sampling [24].

### 2.2. Pretreatment of Wastewater Samples

Immediately after transportation to the laboratory, infectious agents in wastewater are inactivated at 60 °C for 90 min [29,46]. In experiments using surrogate viruses, no loss of RNA yield was noticed with treatment for 30 min at 56 °C [51]. It was observed that wastewater RNA is stable when stored at −20 °C or −75 °C for four weeks, in comparison to 4 °C storage, when a mild decay of RNA was observed [53]. A composite sample of wastewater is a mixture of chemical and biological compounds, insoluble particles, and debris. Before the extraction of nucleic acids, it is vital to filter out components that might hamper subsequent analysis. The majority of the research groups use pre-centrifugation or pre-filtration through various pore sizes (10 μm to 0.2 μm) to remove rough particles and debris [11,19,20,21,22,23,27,28,29,33,39,43,46,47,48]. For the concentration of the sample, two main methods were used: centrifugal or vacuum pump ultrafiltration and the PEG/NaCl precipitation method. The volume chosen for the concentration of the virus should be carefully evaluated based on the actual epidemiological situation. Regions with low prevalence need a higher volume of samples. This, however, leads to concentrating unwanted substances that may hinder further analysis [51]. PEG/NaCl and ultrafiltration were compared by Dumke et al., and significantly higher recovery was measured with the PEG/NaCl precipitation method based on RT-qPCR of the E and S genes [54]. Even better recovery was obtained by Mondal et al. [37]; viral particles captured on silica resin outperformed the PEG/NaCl method. Furthermore, the use of proteases and DNAses seems to have a beneficial effect on releasing viral particles from solid matrices [37]. Similarly, two groups used a glycine buffer to detach virions from organic matter [35,36]. Ultrafiltration was carried out with 10–100 kDa filters, the selection of which should be based on the following downstream processes [11,19,21,22,23,28,33,39,43]. Four studies used electronegative membranes for filtration [21,24,30,38]. Four other research groups adapted the aluminium-based adsorption precipitation method and 5 μm polyvinylidene fluoride beads, respectively [19,20,25,45]. Four groups adapted ultracentrifugation and subsequent analysis of pelleted material [29,40,42,44]. Several studies tried to analyze both components after centrifugation—wastewater supernatants and pellets. In this case, the pellet was homogenized using beads and processed for nucleic acid extraction [22,39,42]. This was performed to evaluate the absorbing capacity of the solid particles present in the wastewater to attract viral fragments. It was investigated whether the concentration of SARS-CoV-2 in sewage sludge and wastewater from the same WWTP could differ by two to three orders of magnitude, as SARS-CoV-2 attaches to solid matrices [51]. On the contrary, the virus accumulated over time in the sludge and therefore might not represent the current state of viral spread in a population.

### 2.3. Extraction of RNA

Nucleic acid extraction is a necessary step in the process of sequencing. High recovery and purity of viral RNA are important parameters to look for when choosing an extraction method. The majority of the methods included here are based on column extraction [19,20,21,22,24,27,29,30,31,34,35,36,37,39,40,41,42,43,44,45,46,47,48]. Column extraction is a feasible, rapid way to extract nucleic acids with high purity. The downside of the method is a limiting size of fragments that attach to the column, depending on the manufacturer, and the potential for high contamination of nucleic DNA [51]. The most used kits were the QiaAmp Viral Mini Kit (Qiagen) or the RNeasy Mini Kit (Qiagen). TRIzol extraction is a cheap elementary method with a high yield of total RNA which also recovers tiny RNA fragments [24,26,28,36,47]. However, the method is time-consuming, prone to pipetting errors, and residual traces of TRIzol or isopropanol may inhibit the PCR reaction when not washed properly [55]. To reduce the concentration of PCR-inhibiting substances, the eluents can be purified by precipitations with alcohol or using a PCR inhibitor removal kit [54,56]. Standard TRIzol extraction was an approach used by two groups, while other groups used a combination of TRIzol extraction with column extraction or magnetic extraction [24,26,28,36]. Three groups adapted magnetic extraction [11,33,38]. The concept of the method is an attachment of negatively charged nucleic acids to positively charged magnetic beads in a magnetic rack with rounds of washes resulting in high purity of extracted deoxyribonucleic acid (DNA) or RNA [57]. Crits-Christoph et al. [19] compared different extraction methods using a commercial kit, silica columns, and the “milk of silica” method. “Milk of silica” is a method developed for direct SARS-CoV-2 RNA extraction from wastewater using 4S, which stands for sewage, salt, silica, and SARS-CoV-2. The method relies on salt, dry silica, and centrifugation, making it very economical. The process takes approximately six hours [19]. Only one study stated that extracted RNA was frozen before sequencing, which can influence the quality of RNA and sequencing outcomes [20].

### 2.4. Confirmation by RT-qPCR

Confirmation of extracted SARS-CoV-2 RNA by RT-qPCR was obtained in 23 out of 31 included studies. Taking all the studies together, at least seven genes were selected as targets to confirm the presence of the virus. The most frequently used were N genes (N1, N2, N3), E, S, ORF1ab, and RdRp genes (Table 2) [19,22,23,24,25,28,29,30,33,34,35,36,37,38,39,40,41,42,44,45,46,47,50]. CDC recommends N1 and N2 targets, while the RdRP gene and confirmatory E gene are based on a recommendation from the Charité Berlin hospital [49,58]. Its worth mentioning that the assays were developed for clinical settings, not for environmental applications [51]. Besides SARS-CoV-2-specific genes, some studies spiked wastewater samples prior to concentration with various, mostly animal, coronaviruses in order to quantify processing and viral extraction efficiency. Moreover, human fecal viruses, such as CrAssphage, JC (John Cunningham) polyomavirus, and Pepper mild mottle virus, were used as markers of human fecal abundance in wastewater [29,31,32,33,35,37,38,39,41,43,45]. Controls, along with a negative control, should be introduced to every assay, as wastewater contains complex matrices. Two studies explicitly stated CT (cycle threshold) cutoff values of less than 36 and one below 40, respectively, to proceed to sequencing [7,11,29]. In addition to RT-qPCR, presence of SARS-CoV-2 was confirmed by digital droplet PCR [40,45] and nested PCR [23,32]. To monitor and avoid inhibition of PCR reactions by inhibitors, two studies introduced the dilution method. Carcereny et al. [41] included duplicate wells containing undiluted RNA and duplicate wells containing a 10-fold dilution to monitor the presence of inhibitors [41]. Rios et al. [42] diluted samples to gain a 50–150 ng/reaction, and in case of a concentration below 50 ng, the sample was diluted two-fold to dilute matrix-related inhibitors [42]. As can be seen in Table 1, there is high variability in the target genes used with PCR master mixes; therefore, unification for comparison studies would be beneficial.

### 2.5. Sequencing Technology

Sequencing was carried out using different technologies, but Illumina sequencing technology and Oxford Nanopore Technology (ONT) were used in most cases. Both technologies are comparable with respect to the identification of SARS-CoV-2 and its variants; however, slight differences exist. Illumina sequencing is next-generation sequencing; analysis is not available in real-time, optimal reads are shorter, around 150 bp, with a high accuracy of around 99%. Furthermore, starting and reaction chemistry costs are substantially higher in comparison with ONT. ONT uses amplicon sequencing with lower accuracy (92–97%), has a high throughput, the reads can be extremely long, and it is possible to carry out real-time analysis [59,60]. It is known that because of the hostile wastewater environment, RNA is fragmented, and shorter fragments are present [61]. The sequencing library was prepared with multiple different panels or primer sets. Sixteen groups rely on primers sets covering nearly the whole genome of SARS-CoV-2, from which 12 groups used ARTIC primers designed by the ARTIC network [11,20,21,25,29,33,38,40,42,43,44,46]. The ARTIC network is an international cooperation of research universities which aims to develop an end-to-end system for controlling viral epidemics and gathers information used by public health bodies. Two sets of primers are provided for Illumina with average lengths of 150 and 400 bp, while ONT uses only the latter. The coverage of the primers is over 98% of the SARS-CoV-2 genome. Five groups used the Swift Nomalase Amplicon SARS-CoV-2 panel, which covers 98%, using 341 amplicons, and once an improved version called the Swift Nomalase Amplicon SARS-CoV-2 Additional Genome Coverage panel was used, which covers 99.7%, with 345 amplicons. The average size of the amplicons is 150 base pairs and they are designed exclusively for Illumina platforms [27,28,36,37,47]. In one case, the research group designed two pools of 89,500 bp primers [11]. Several groups analyzed only the presence of specific genes for detection of the virus and mutations of interest at these genome sites [19,24,26,30,31,32,34,35,39,45]. The SARS-CoV-2 isolate Wuhan-Hu-1, with the accession number NC_045512 or MN908947, was used by all groups as a reference genome for genomic analysis.

### 2.6. Data Analysis and Variant Identification

Detection of SARS-CoV-2 in wastewater has its limitations compared to clinical sampling. The major obstacle is that amplicons are from a mixed pool of individuals and thus it is unlikely that mutations in different amplicons or even in the same amplicon can be associated with a single genome or variant. However, signature mutations associated with specific variants can still be detected and thus complement clinical datasets, which will always be limited by the number of patients tested, most of whom are symptomatic [53].

Several combinations of tools were used to carry out data analysis and variant identification in the presented studies, and no single dominant approach was observed. The optimal procedure for each research unit would be to test a benchmark dataset with multiple tools for a given task and evaluate their performance. Therefore, we focus on data analysis used with ARTIC primer sets, which were used in 11 out of 31 studies. To align generated reads to a reference genome, minimap2, the Burrows–Wheeler aligner (BWA), or medaka are good choices, and these were used in several studies [20,38,40,43]. GATK, samtools, iVAR, freebayes, or Octopus can be employed to call single nucleotide polymorphisms (SNPs) and short indels. These can be compared with the list of known SARS-CoV-2 variants. These tools were implemented in at least five studies that used ARTIC primer sets [20,25,38,40,42].

The methods used for the identification of variants varied between the assessed studies. Simple approaches include the subjective association of identified mutations with major variants of interest, often focusing only on the identification of one prevalent variant of interest [11,24,38,44]. Another approach utilizes diverse deconvolution steps for the assignment of SARS-CoV-2 lineage based on the frequencies of identified mutations in a dataset. These are typically implemented using custom-designed scripts and pipelines [20,21,42,43,62]. Alternatively, consensus sequences can be prepared (for example, in medaka [63]. These are used by the Pangolin tool [64] for phylogenetic assignment, as in the paper by Rios et al. ([42]). However, creating reliable SARS-CoV-2 consensus sequences can be problematic, since the consensus sequence generated from a WW sample is presumably representative of either a dominant strain or a composite of multiple strains circulating at the time of sampling ([43]). An overview of a standard pipeline is visualized in Figure 2. The left side of the flowchart (Figure 2) represents the general processing of sequenced wastewater samples, with examples of the tools used on the right.

## 3. Discussion

Based on the rich biochemical composition of wastewater samples, viral concentration and extraction are critical factors in obtaining the amounts of highly preserved viral particles required for successful sequencing [11]. Generally, the obtainment of particles was accomplished in the studies by running pre-sequencing RT-qPCR, where CT < 36 was necessary for obtaining a near-complete genome of the virus [11,19,27]. Sampling of wastewater also avoids many of the disadvantages of clinical sampling. It is easily collected, there are few if any problems regarding ethical issues, and information can be acquired about current epidemic situations based on large populations. However, it has its limits, such as fragmented virion particles in low titers and exposure to harmful wastewater ecosystems [52,65] detected one infected person from 60 residents by analyzing wastewater with RT-qPCR [52]. In Japan, scientists reached an even higher sensitivity, detecting one infected individual in 100,000 by RT-qPCR [66]. However, sequencing of wastewater for the re-assembly of a whole viral genome will need highly preserved RNA.

Therefore, optimizing and creating an ideal workflow for preserving viral particles, or at least unfragmented RNA molecules, is crucial for the process. Using a refrigerated sampling system for composite samples and keeping a sample on ice until analysis is vital for inhibiting damaging enzymes, such as RNAses [53]. The selection of pretreatment methods is foremost based on the following analysis and desired outcome. In short, the best option is to begin with particle release by enzymes or buffers, followed by rough filtration to prevent the clogging of the fine ultrafilters with pore sizes of 10–100 kDa. Another commonly used method is the PEG 8000/NaCl precipitation method. Extraction of RNA is the last step before sequencing. The extraction can be performed by various methods with different advantages. Basic TRIzol extraction is suitable for capturing total RNA, including tiny fragments. Column-based extractions are fast, yielding high concentrations of RNA with high purity. When using this method, the minimal size of fragments captured by the column needs to be considered to avoid the loss of significant portions of RNA [67]. Magnetic bead extraction yields an even higher purity of RNA. Unfortunately, this method is the least economical because of the expensiveness of the beads and its lack of a sufficient throughput. Silica resin capture of viral or nucleic acid particles might be potentially very effective, but more studies are needed for confirmation. In this review, we have presented the two most frequently used sequencing technologies: Illumina and ONT. While both are sufficient for monitoring SARS-CoV-2 variants, Illumina, with its higher accuracy, could be used for novel mutation discoveries. On the other hand, ONT is more affordable and its high throughput is crucial for assessing current epidemic situations.

Several studies included in this review detected variants that were not detected in clinical testing at the current geographical location, while Peréz-Cataluna et al. [25] identified three novel variants in the spike gene in a comparison with the Wuhan-Hu-1 isolate (MN908947.3) [19,25,26]. Similar to monitoring through RT-qPCR, sequencing of wastewater could potentially yield predictions of the epidemic situation of SARS-CoV-2 in the human and, potentially, animal/livestock populations of a given geographical area [11]. The early detection of mutations with predispositions for higher transmissibility would offer a chance for corrections to mathematical models of disease spread, and this information could be used by public authorities to modify health restrictions. This would be especially useful in refugee camps, elderly residences, and medical facilities [51]. Wastewater samples would also allow for a retrospective view of the evolution of pandemics if correctly stored. With this approach, we can make a map of mutations across a country showing the predominant variants. Knowledge of circulating mutations can be used to design new, more effective vaccines, as with flu vaccines [23]. This is even more important when vaccination and convalescent plasma treatment are broadly available and, together with natural infection, the biological pressure on the virus to mutate will increase [68]. Sequencing of wastewater can be systematically implemented in the current monitoring of SARS-CoV-2. Compared with RT-qPCR, genome sequencing does not rely on a specific pair of primers and a probe; therefore, introducing a specific mutation will not lower the method’s sensitivity [19].

The importance of the technique in regular testing will rely on creating standardized protocols for sampling, pretreatment, concentration, and extraction of samples, including sequencing [54,69]. Currently, there are substantial differences between methods in terms of recovery of the virus and its surrogates, with values ranging from 3.3–73% [51]. Nowadays, many researchers are looking for an optimal solution to the presented challenges. After establishing standardized protocols, it will be possible to analyze microbial as well as viral presence in wastewater, as is already done in the case of drugs, alcohol, pharmaceuticals, pesticides, and other harmful agents [70,71]. Another limitation of the method is that it is not possible to analyze the age cohort distribution of infected individuals, and the analyzed area is definite to the working sewage system. Analysis can be further hindered by a large share of industrial wastewater (e.g., pharmaceuticals), retention time in sewage, and the definition of the number of connected populations. It is necessary to investigate how the virus behaves in wastewater and how microbial composition, chemical pollution, type of sewage, temperature, and pH affect the virus. For a reliable methodology, it is also crucial to investigate shedding of the virus in feces during infection.

As the SARS-CoV-2 pandemic has extensive social and economic consequences worldwide, it is in the interest of national bodies to realize the potential assets of wastewater monitoring and epidemiology. For example, in Europe, there is already a working network of research institutes monitoring the use of illegal drugs which publishes their data at a transnational level [72,73,74]. In addition to this, the NORMAN initiative is already trying to unify the whole process of RNA SARS-CoV-2 monitoring in wastewaters at the European level [75]. The European commission started developing a HERA incubator at the beginning of 2021 for bio-defense preparedness with a specific subtask of monitoring wastewaters for RNA SARS-CoV-2 presence. This could lead to other scientific developments for wastewater monitoring, such as microsensors, data collection, and artificial intelligence [76]. Based on quantitative measurement of specific biomarkers, drugs, metabolites, and viruses in wastewaters from different cities and locations, we can evaluate the lifestyle of, the occurrence of diseases in, or the impact of the environment on a given population. Therefore, analysis of wastewaters can be equivalent to urine or blood analysis of a population from a specific area [70].

## 4. Conclusions

In this review, we have tried to describe the pros and cons of different pipelines and elucidate the current state of wastewater-based epidemiology via sequencing. In summary, based on the literature, the best option is to collect a 24 h composite sample, which should be kept on ice until reaching the laboratory. After deactivation of the virus, large wastewater particles should be depleted, followed by PEG precipitation or concentration by ultrafiltration and TRIzol or column-based RNA extraction. PCR inhibitors should be removed to ensure purity to gain the lowest possible CT value for the chosen genes for SARS-CoV-2. Samples with CT values below 36 should be used further for sequencing according to ARTIC network recommendations. Sequencing technology should be chosen based on the individual capabilities of laboratories and the desired outcomes. Variations in each step of processing are possible. However, the standardization of procedures is greatly needed to ensure the comparability of wastewater studies and eliminate biased results.

## Figures and Tables

**Figure 1 ijerph-19-09749-f001:**
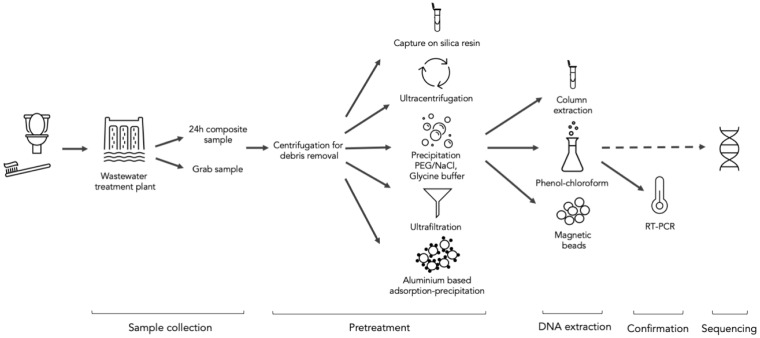
Scheme describing step-by-step methodologies for SARS-CoV-2 sequencing from wastewater.

**Figure 2 ijerph-19-09749-f002:**
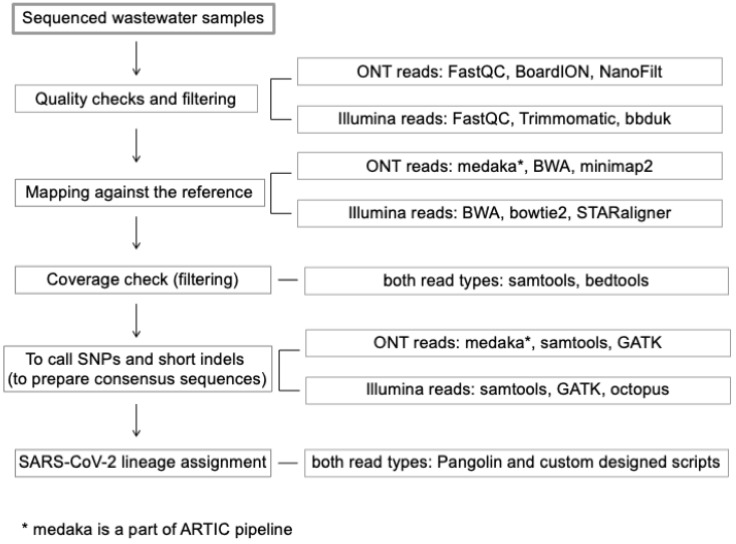
Scheme describing the data analysis and variant identification pipeline, with examples of frequently used tools.

**Table 1 ijerph-19-09749-t001:** Literature review. Information about sampling, pretreatment, extraction of RNA, RT-qPCR, and sequencing technology for the detection SARS-CoV-2 in wastewater.

Extraction	RT-qPCR Confirmation Genes	PCR Kit	Seq Technology	Primers	Ref. Genome	Reference
Magnetic extraction with NucliSENS easyMAG system (Biomerieux, Marcy-l’Étoile, France)	N1–N3 and E	Not specified	Illumina (San Diego, CA, USA)/ONT MinION (Oxford, UK)	2 pools of 89,500 bp primers	2 datasets—Dutch–Belgian full length SARS-CoV-2 genomes (8th of July 2020) and GISAID database (1st of July 2020)	[11]
QiaAmp Viral MiniKit (Qiagen, (Hilden, Germany))	None	None	Illumina NovaSeq 6000 (San Diego, CA, USA)	ARTIC V3 primers	GISAID	[20]
AllPrep DNA/RNA Mini Kit (Qiagen (Hilden, Germany))	N1	TaqPath 1-Step RT-qPCR Master Mix or TaqMan Fast Virus 1-Step Master Mix (Thermo Fisher Scientific (Waltham, MA, USA))	Illumina NextSeq 550 (San Diego, CA, USA)	Respiratory Virus Oligo Panel and Gut Microbiome	EPI_ISL_402124 and GISAID (23 August 2020)	[19]
Direct extraction of sample with silica columns (Zymo III-P (Irvine, CA, USA))	None					
“Milk of Silica” method	None					
RNeasy PowerMicrobiome Kit+ Onestep PCR inhibitor removal kit (Zymo Research (Irvine, CA, USA)) following concentration step	N2	Not specified	Illumina MiSeq (San Diego, CA, USA)	ARTIC V3 primers	NC_045512	[21]
RNeasy Mini Kit (Qiagen, (Hilden, Germany))	N1 and N2	2019-nCoV CDC RUO Kit, TaqPath 1-Step RT-qPCR Master Mix, CG (Thermo Fisher Scientific (Waltham, MA, USA))	ONT MinION (Oxford, UK)	ARTIC V3 primers	MN908947.3	[22]
RNeasy Mini Kit (Qiagen (Hilden, Germany)						
High Pure Viral RNA kit (Roche (Basel, Switzerland)	RdRp, ORF8b, and specific primers for detection of B.1.1.7 variant	Not specified	Illumina MiSeq (San Diego, CA, USA)	Not specified	GISAID (1st of March 2021)	[23]
Direct-Zol RNA Miniprep kit (Zymoresearch (Irvine, CA, USA)) or Trizol Reagent (Invitrogen (Waltham, MA, USA))	N genes	nCov CDC EUA Kit	ONT MinION (Oxford, UK)	13 sets of primers covering N gene, S gene, ORF1ab, and ORF8	NC_045512	[24]
Maxwell RSC Pure Food GMO and Authentication Kit (Promega (Madison, WI, USA)	N1, E, and RdRp	RT-qPCR using One Step PrimeScriptTM RT-PCR Kit (Perfect Real Time) (Takara Bio (Kusacu, Japan))	Illumina MiSeq (San Diego, CA, USA)	ARTIC V3 primers	MN908947.3	[25]
TRIzol extraction	None	None	Illumina iSeq100 (San Diego, CA, USA)	Primers targeting RBD domain	MN908947	[26]
RNaeasy MiniKit (Qiagen (Hilden, Germany))	E	SuperScript III Platinum One-Step qRT-PCR Kit (Invitrogen (Waltham, MA, USA))	Illumina HiSeq 2500 (San Diego, CA, USA)	Swift Nomalase Amplicon SARS-CoV-2 panel (SWIFT (Coralville, IA, USA))	NC_045512.2	[27]
TRIzol and DNA-zol 96 MagBead RNA kit (Zymoresearch (Irvine, CA, USA))	None	Taq 1-Step Multiplex Master Mix (Thermo Fisher (Waltham, MA, USA))	Illumina (San Diego, CA, USA)	Swift Nomalase Amplicon SARS-CoV-2 panel (SWIFT (Coralville, IA, USA))	NC_045512.2	[28]
QiaAmp Viral MiniKit (Qiagen (Hilden, Germany))	N2	SuperScriptTM III PlatinumTM One-Step qRT-PCR Kit (Invitrogen (Waltham, MA, USA))	Illumina MiSeq (San Diego, CA, USA)	Primers targeting whole SARS-CoV-2 genome	Not specified	[29]
RNeasy PowerMicrobiome Kit (Qiagen (Hilden, Germany))	N1, N2, E	iTaq™ Universal Probes One-Step Reaction Mix (Bio-Rad Laboratories (Hercules, Clearwater, FL, USA)	Illumina MiSeq (San Diego, CA, USA)	Six N1 primers	NC_045512.2	[30]
QiaAmp Viral MiniKit (Qiagen (Hilden, Germany))	None	None	Illumina MiSeq (San Diego, CA, USA)	Primers covering 3 regions of S gene	NC_045512.2	[31]
Water DNA/RNA magnetic bead kit (Idexx (Westbrook, ME, USA)	None	Novel nested and nested real-time PCR, dNTPs mix (Jena Bioscience GmbH (Jena, Germany)), Kapa Taq polymerase (Kapa Biosystems (Potters Bar, UK); for probe fluorescent-based real-time PCR Kapa Probe Fast Universal (2X) qPCR Master Mix (Kapa Biosystems (Potters Bar, UK)	Semi-conductor sequencing technology	In-house design primers for S gene and 5 specific regions	NC_045512.2	[32]
Magnetic extraction with NucliSENS easyMAG system (Biomerieux (Marcy-l’Étoile, France))	N1 and E	1 × RNA Ultrasense Reaction Mix with 1 µL RNA Ultrasense Enzyme Mix (Invitrogen (Waltham, MA, USA)	Illumina MiSeq (San Diego, CA, USA)	ARTIC V3 primers	MN908947.3	[33]
Spin star nucleic acid kit (BioservUK (Rotherham, UK))	ORF1ab, N, and E	(Kit 1) Real-Time Fluorescent RT-PCR Kit for detecting 2019-nCoV by BGI China (Shenzhen, China) (IVD and CE marked; catalogue no. MFG030010), takes ORF 1ab gene as the target domain, (Kit 2) qRT-PCR for Novel Coronavirus (2019-nCoV) Nucleic Acid Diagnostic Kit (PCR-Flourescence, Probing, IVD marked) by Sansure Biotech (Sansure Biotech Inc China, ref no. S3102E, Changsha, China). The kit utilizes the novel coronavirus (2019-nCoV) ORF-1 gene and a conserved coding nucleocapsid protein N-gene as the target regions and finally (Kit 3) detection Kit for 2019 Novel Coronavirus RNA (PCR-Fluorescence Probing), for the E gene: Superscript III one step RT-PCR with platinum Taq Polymerase (Invitrogen (Waltham, MA, USA))	ABI 3100 (Thermofisher (Waltham, MA, USA))	ORF1a primer	Not specified	[34]
QiaAmp Viral MiniKit (Qiagen (Hilden, Germany))	S, N1, and RdRP	OneStep qPCR Quantinova kit (Qiagen (Hilden, Germany))	Illumina MiSeq (San Diego, CA, USA)	S and RdRP primers	NC_045512.2 and other coronaviruses	[35]
TRIzol extraction and QiaAmp Viral MiniKit (Qiagen (Hilden, Germany))	N1	4× TaqMan Fast Viral One-Step Master Mix (Thermo Fisher (Waltham, MA, USA))	Illumina MiSeq (San Diego, CA, USA)	Swift Nomalase Amplicon SARS-CoV-2 panel (SWIFT (Coralville, IA, USA))	NC_045512.2	[36]
Maxwell RSC Pure Food GMO and Authentication Kit (Promega (Madison, WI, USA)), Maxwell Enviro Wastewater kit (Promega (Madison, WI, USA)), Wizard Enviro Wastewater TNA kit (Promega (Madison, WI, USA))	N1, N2, and E	SARS-CoV-2 RT-qPCR Detection Kit for Wastewater (Promega Corp. (Madison, WI, USA))	Illumina MiniSeq (San Diego, CA, USA)	Swift Nomalase Amplicon SARS-CoV-2 Additional Genome Coverage panel (SWIFT (Coralville, IA, USA))	NC_045512.2	[37]
Magnetic extraction with NucliSENS easyMAG system (Biomerieux (Marcy-l’Étoile, France))	E	SensiFast reaction mix (Bioline (London, UK))	Illumina MiSeq (San Diego, CA, USA)	ARTIC V3 primers	NC_045512.2	[38]
QiaAmp Viral MiniKit (Qiagen (Hilden, Germany)))	N1, N2	Allplex™ 2019-nCoV Assay (Seegene (Wallnut Creek, USA)	Illumina MiSeq (San Diego, CA, USA)	VirCapSeq Enrichment kit (Roche (Basel, Switzerland))	NC_045512.2	[39]
AllPrep PowerViral DNA/RNA kit and Rneasy Power Microbiome kit (Qiagen (Hilden, Germany)))	N1	One-step QuantiNova Multiplex RT-PCR Kit	Illumina HiSeq 1500 (San Diego, CA, USA)	ARTIC V2 primers	NC_045512.2	[40]
Maxwell RSC Pure Food GMO and Authentication Kit (Promega (Madison, WI, USA))) and NucleoSpin RNA Virus kit (Macharey-Nagel (Duren, Germany))	N1, S	N1: PrimeScript One-Step RT-PCR Kit (Takara Bio (Kusacu, Japan)) and 2019-nCoV RUO qPCR Probe Assay primer/probe mix (IDT, Integrated DNA Technologies (Newark, NJ, USA)); S:TaqMan RT-qPCR assay and RT-qPCR mastermix was prepared using the PrimeScript One-Step RT-PCR Kit (Takara Bio (Kusacu, Japan))	Illumina MiSeq (San Diego, CA, USA)	ARTIC V3 primers (S gene only)	NC_045512.2	[41]
AllPrep PowerViral DNA/RNA kit and Rneasy Power Microbiome kit (Qiagen (Hilden, Germany)))	N1	qScript XLT One-Step RT-qPCR ToughMix (Quantabio (Beverly, MA, USA))	ONT PromethION (Oxford, UK)	ARTIC V3 primers	MN908947	[42]
AllPrep PowerViral DNA/RNA kit (Qiagen (Hilden, Germany)))	None	None	ONT GridION X5 (Oxford, UK)	ARTIC V3 primers	MN908947.3	[43]
EZ1 Virus Mini Kit (Qiagen (Hilden, Germany)))	N genes	Superscript III Platinum One-step Quantitative RT-PCR systems with the ROX kit (Invitrogen (Waltham, MA, USA))	Illumina NovaSeq 6000 (San Diego, CA, USA)	ARTIC V3 primers	NC_045512.2	[44]
RNeasy PowerMicrobiome Kit (Qiagen (Hilden, Germany)))	N1, N2, and E	High-Capacity cDNA Reverse Transcription Kit (Applied Biosystems (Waltham, MA, USA)), ddPCR	Illumina NextSeq 550 (San Diego, CA, USA)	COVIDSeq kit (Illumina (San Diego, USA))	Not specified	[45]
Rneasy Power Water Kit (Qiagen (Hilden, Germany)))	E, RdRP, N	TRUPCR SARS-CoV-2 RT qPCR kit (V-3.2) (3B BlackBio Biotech India Limited (Bhopal, India))	ONT MinION (Oxford, UK)	ARTIC V3 primers	MN908947.3	[46]
Trizol Reagent (Invitrogen (Waltham, MA, USA)) and Direct-Zol RNA Miniprep kit (Zymoresearch (Irvine, CA, USA)) and RNA cleanud up with Zymo RNA Clean & Concentrator (Zymoresearch (Irvine, CA, USA))	N1	Taq 1-Step Multiplex Master Mix (Thermo Fisher Scientific (Waltham, MA, USA))	Illumina NextSeq 550 (San Diego, CA, USA)	Swift Nomalase Amplicon SARS-CoV-2 panel (SWIFT (Coralville, IA, USA))	NC_045512.2	[47]
QiaAmp Viral MiniKit (Qiagen (Hilden, Germany)))	None	None	Illumina NextSeq 500 (San Diego, CA, USA)	CleanPlex SARS-CoV-2 FLEX panel (Paragon Genomics)	NC_045512.2	[48]

**Table 2 ijerph-19-09749-t002:** Recommended genes for SARS-CoV-2 RT-qPCR assays. (Adapted from WHO.)

Institution	Target Genes
US CDC, Atlanta, GA, USA	N1, N2, N3
Charité, Berlin, Germany	RdRp, E, N
China CDC, Beijing, China	ORF1ab and N
Institute Pasteur, Paris, France	Two targets in RdRp (IP2 and IP4)
National Institute of Infectious Diseases, Tokyo, Japan	Pancorona and multiple targets, Spike protein
HKU, Hong Kong, China	ORF1b-nsp14, N
National Institute of Health, Nonthaburi, Thailand	N

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
