# Peer review of "Wastewater Sequencing—An Innovative Method for Variant Monitoring of SARS-CoV-2 in Populations"

_ijerph, 2022, doi:10.3390/ijerph19159749_

Round 1
Reviewer 1 Report
Dear Author
Excellent summary and description of the different methods of extraction and rt-pcr, a little less for sequencing, where I would move the different methods used
I simply asked in paragraph 2.5. Sequencing technology describe other sequencing methods such as ABI3100.
Author Response
Dear reviewer,
thank you for your comments.
"I simply asked in paragraph 2.5. Sequencing technology describe other sequencing methods such as ABI3100."
I am sorry, but we can not find ABI3100 sequencing method used in SARS-CoV-2 and wastewater therefore we did not broaden the sequencing section. If you will be that kind and let us know about the publication that mentioned ABI3100 and the detection of SARS-CoV-2 in wastewater we would highly appreciate it.
Thank you.
Reviewer 2 Report
The manuscript, a review, entitled “Wastewater sequencing – an innovative method for variant monitoring of SARS-CoV-2 in population” by Tamas et al provides an overview of the methods that are currently being used to detect/monitor SARS-CoV-2 in waste waters. The review is well written and describes in detail various steps involved from sample preparation to sequencing of viral RNAs.
The review can be published as is except for the very minor suggestions below.
Page 10, line: 221, Describe the full form of CT (cycle threshold), when it is mentioned at the first instance.
Page 13, line: 329, “clotting”, shouldn’t it be “clogging” instead
Page 15, line: 401, Consider replacing “With this review, we tried to propose ….” with “In this review, we tried to describe ….”
Author Response
Dear reviewer,
thank you for your comments.
We corrected the minor spellings and errors according to your suggestions.
All the changes can be seen already in the document.
Thank you.